# Molecular Mechanisms and Animal Models of HBV-Related Hepatocellular Carcinoma: With Emphasis on Metastatic Tumor Antigen 1

**DOI:** 10.3390/ijms22179380

**Published:** 2021-08-29

**Authors:** Yung-Tsung Li, Hui-Lin Wu, Chun-Jen Liu

**Affiliations:** 1Department of Internal Medicine, College of Medicine, National Taiwan University, Taipei 100, Taiwan; yungtsungli@gmail.com; 2Graduate Institute of Clinical Medicine, College of Medicine, National Taiwan University, Taipei 100, Taiwan; 3Hepatitis Research Center, National Taiwan University Hospital, Taipei 100, Taiwan

**Keywords:** hepatitis B virus, chronic hepatitis, hepatocarcinogenesis, biomarker, splice variant, MTA1, oncogene, transgenic mice, hydrodynamic injection

## Abstract

Hepatocellular carcinoma (HCC) is an important cause of cancer death worldwide, and hepatitis B virus (HBV) infection is a major etiology, particularly in the Asia-Pacific region. Lack of sensitive biomarkers for early diagnosis of HCC and lack of effective therapeutics for patients with advanced HCC are the main reasons for high HCC mortality; these clinical needs are linked to the molecular heterogeneity of hepatocarcinogenesis. Animal models are the basis of preclinical and translational research in HBV-related HCC (HBV-HCC). Recent advances in methodology have allowed the development of several animal models to address various aspects of chronic liver disease, including HCC, which HBV causes in humans. Currently, multiple HBV-HCC animal models, including conventional, hydrodynamics-transfection-based, viral vector-mediated transgenic, and xenograft mice models, as well as the hepadnavirus-infected tree shrew and woodchuck models, are available. This review provides an overview of molecular mechanisms and animal models of HBV-HCC. Additionally, the metastatic tumor antigen 1 (MTA1), a cancer-promoting molecule, was introduced as an example to address the importance of a suitable animal model for studying HBV-related hepatocarcinogenesis.

## 1. Introduction

Liver cancer exhibits high mortality. It is the seventh most commonly diagnosed cancer but ranks as the third leading cause of cancer-related death. In 2020, there were more than 905,677 newly diagnosed primary liver cancer cases and 830,180 deaths worldwide [1]. The high mortality rate of liver cancers is mainly due to the lack of sensitive biomarkers for early diagnosis and effective therapeutics for patients with advanced hepatocellular carcinoma (HCC), as well as the frequent occurrence of relapse [2]. HCC is the most common type of liver cancer in adults and accounted for approximately 90% of cases [3]. In hepatitis B virus (HBV) highly endemic areas, including Taiwan, chronic hepatitis B infection is the most common cause of HCC [4,5,6] and accounted for approximately 75% to 90% of all HCC cases, according to the updated statistics [7,8]. Specific viral factors (e.g., the high HBV viral load [9,10], HBV genotypes [11,12,13,14], HBV genomic mutations [15] and viral oncogenic proteins), host factors (e.g., increasing age, male gender, African/Asian race) [16], and even unhealthy lifestyles have been proven to contribute to an increased risk of HBV-related HCC (HBV-HCC) [13,17,18]. Furthermore, various external cofactors have also been shown to cooperate with HBV to promote HCC development [7,19].

HBV is a small DNA virus that belongs to the family *Hepadnaviridae* and predominantly infects the liver. To date, 10 genotypes (A–J) have been identified according to the differences in the whole HBV genome sequence [20]. The HBV genome is a 3.2 kb partially double-stranded DNA containing four overlapping open reading frames (ORFs), which are designated as C, S, P, and X genes and encode for core/HBeAg, surface, polymerase, and X proteins, respectively (Figure 1). It has been confirmed that HBV can promote hepatocarcinogenesis through a variety of mechanisms, which ultimately lead to the final pathogenic outcomes.

Vaccination and anti-viral therapies to inhibit new infections and viral replication of HBV is the first step in preventing the occurrence of HBV-HCC. These strategies are highly efficacious in terms of interrupting the progression from chronic infection to HCC but do not eliminate the risk of HCC development entirely [21,22,23]. Therefore, early detection of HCC in chronic hepatitis B (CHB) patients, effective treatments, and prevention of recurrence are of great importance for further reducing HBV infection-associated mortality. Currently, several predicting models incorporating clinical features and viral factors (e.g., serum levels of HBV DNA, HBeAg, and HBsAg) have been developed to evaluate the risk of HCC development in CHB patients [21], with periodic screening using ultrasonography with alpha-fetoprotein (AFP) among high-risk groups the most frequently used strategy for early detection of HBV-HCC. However, the sensitivity and specificity of these methods are not satisfactory so far. Surgical resection is considered the first-line therapeutic option for HCC in the initial phase; nevertheless, only a few patients are candidates for surgery [24,25] because most patients are diagnosed at the advanced stage, where few therapeutic options are available. Although the advent of molecular-targeted drugs, such as the multi-kinase inhibitor Sorafenib, has advanced the systemic therapies for advanced-stage patients, the survival benefit remains quite limited [26,27], and the long-term use of these drugs is frequently associated with toxicity and drug resistance [28,29]. Furthermore, even in patients with resectable tumors, frequent tumor recurrence after surgery remains the major cause of death for HBV-HCC. The heterogeneous molecular nature of HCC affecting treatment outcome and recurrence is another reason for the poor prognosis of HCC [4]. These issues highlight the need to develop more effective approaches for early detection, recurrence prediction, and treatment of advanced HBV-HCC. Advancements in biomedical research and technologies have greatly facilitated the identification of more gene expression signatures and molecular mechanisms associated with HBV-HCC and made great progress in many aspects. Novel therapeutic options, such as different molecular targeted therapies, epigenetic modulators, immunotherapies, oncolytic virotherapy, and even chemopreventive agents, alone or in combination, are continuously emerging [30]. New biomarkers for early detection and prediction of treatment outcomes are also under active investigation [31]. Employment of appropriate HBV-HCC animal models that are relevant to human clinical settings is of paramount importance to successfully translate basic scientific findings into clinical applications for patients.

In this review, we first overviewed the current knowledge regarding the molecular mechanisms of HBV-induced hepatocarcinogenesis. We then discussed the advantages and limits of different HBV-HCC animal models, particularly focusing on their capacity to mirror HBV-induced hepatocarcinogenesis. In the last section, we focused on MTA1 (metastatic tumor antigen 1), a cancer-promoting molecule that we have studied extensively and is particularly overexpressed in HBV-HCC. We summarized the molecular mechanisms of MTA1 in tumorigenesis and metastatic progression of HBV-HCC. To demonstrate the value of animal models in the study of HBV-HCC, we used MTA1 as an example to illustrate how we translated the findings from the woodchuck HCC model into human clinical research and demonstrated the potential clinical application of MTA1 and its major splice variant, MTA1dE4, in predicting early recurrence of HBV-HCC [32,33].

## 2. Oncogenic Mechanisms of HBV

HBV can promote hepatocarcinogenesis through both direct and indirect pathways (Figure 2). There have been numerous studies describing the multiple mechanisms involved in this multistage process. (i) HBV can integrate into the host genome, which leads to genome instability and insertional mutagenesis, resulting in the abnormal expression of oncogenes and tumor suppressor genes. (ii) Viral proteins HBx and HBs can activate endoplasmic reticulum stress- and carcinogenesis-associated signaling pathways through regulating gene expression at multiple steps. A variety of important biological processes, such as proliferation, apoptosis, DNA damage repair, mitochondrial functions, etc., can be altered by the pleiotropic functions of the viral oncoproteins and result in the development of HCC. (iii) Chronic HBV infection can induce the production of proinflammatory cytokines and oxidative stress, create an inflammatory microenvironment, and cause liver injury. Cycles of inflammation, injury, and regeneration lead to the development of HCC.

### 2.1. HBV DNA Integration Promotes the Occurrence of HCC

Although it is not an essential step in the viral life cycle, integrated HBV DNA has been observed early after infection [34,35,36,37,38], throughout chronic infection [39,40,41], and even after viral clearance [37,42]. Furthermore, approximately 80% to 90% of tumor cells from patients with HBV-HCC [43,44] contain integrated HBV DNA. Genomic modeling also supports the notion that HBV integration is common in HBV-HCC patients [45]. It has been shown that HBV insertions are associated with genetic remodeling of the host genome, including the host gene deletions and translocations, the acquisition of viral sequences, and the generation of fusion transcripts [46]. These genetic alternations in the host genome may alter the expression of oncogenes and tumor suppressor genes, all of which may aid in cell transformation and HCC progression.

The tumorigenicity of HBV integration depends, to a certain extent, on the function of HBV integration targeted host genes in HCC. HBV integration usually occurs at random sites in the genome of the host cell, but recently, integration hotspots were identified by using next-generation sequencing. Among them, *TERT* (telomerase reverse transcriptase) is the most frequently integrated gene [43,47,48], followed by *MLL4* (Myeloid/lymphoid or mixed-lineage leukemia 4) [43,49]. *FN1* (Fibronectin 1) [43,50], *CCNE1* (Cyclin E1) [43,51] *CCNA2* (Cyclin A2) [43,51] and *ROCK1* (Rho-associated, coiled-coil containing protein kinase 1) [43,51] have also been reported. (For a more complete list of the integrated genes, please refer to the review of Lee et al. [52].) These targeted genes for HBV integration usually are closely related to tumor progression and also are shown to be significantly enriched in cancer-related pathways [50,53]. In addition, integrated HBV DNA becomes a stable source for the production of truncated or mutated HBx and pre-S1/S2 proteins whose expression may also contribute to hepatocarcinogenesis [54].

### 2.2. HBx Is a Multifunctional Viral Protein with Versatile Oncogenic Activities

The HBx protein is known to play an important role in the HBV life cycle and HBV-induced hepatocarcinogenesis [51]. Accumulating lines of evidence demonstrate that HBx dysregulates the expression of many cellular genes, leading to the transformation of normal cells into tumor cells. Significant biological processes, mainly cell replication control, mitochondrial functions, DNA repair, etc., are modulated by the expression of HBx [55]. Important signaling pathways involved in HCC, such as β-Catenin pathway [56], Src-dependent phosphatidylinositol-3 kinase pathway (PI3K/AKT) pathway [57], Ras-Raf-mitogen activated protein kinase (MAPK) pathway [57], nuclear factor kappa B (NF-κB) pathway [58,59], Janus kinase/STAT (JAK/STAT) pathway [60] and protein kinase C (PKC) pathway, have all been reported to be aberrantly activated by HBx [61,62].

HBx exerts its pleiotropic functions via interacting with a variety of proteins. For a complete list of the HBx interacting proteins, please refer to review papers [63,64]. HBx per se cannot bind directly to DNA; instead, HBx transactivates viral and cellular promoters through interacting with transcription factors, for example, MYC [65], cAMP response element-binding protein (CREB) [66], activating protein 1 (AP-1) [67], RNA polymerase binding protein [68], nuclear factor kappa B (NF-κB) [59], etc. HBx can also mediate epigenetic plasticity by interacting with histone acetyltransferase [69] and core components of methyltransferase complexes and, e.g., DNMT3a, DNMT3b, and WD Repeat Domain 5 Protein [70], etc. Aberrant DNA methylation and histone acetylation induced by HBx results in the activation of oncogenes and the silencing of suppressor genes [71], favoring the development of tumors. Moreover, via interacting with components of the ubiquitination system, especially E3 ubiquitin ligase complexes, HBx can function on the ubiquitin protein degradation system in either a positive or negative manner and affect the fates of many proteins [72,73]. Through the above-mentioned mechanisms, HBx is actively involved in tumorigenesis.

### 2.3. Roles of HBV Surface Protein in HCC

Apart from HBx, HBV surface antigens (HBsAgs), especially the pre-S1 and pre-S2 mutants, have been reported to be associated with hepatocarcinogenesis via endoplasmic reticulum (ER) and non-ER stress-dependent pathways [51,74]. Ground glass hepatocytes (GGHs) that harbor pre-S mutant proteins represent histologically preneoplastic lesions of HCC in patients with chronic HBV infection [75,76]. During HBV infection, the inappropriate overexpression and accumulation of HBsAgs in the ER of the infected hepatocytes causes ER overload and induces ER stress. In addition, the mutation-induced misfolding of pre-S mutants, which is prevalent in patients with chronic hepatitis B and HCC [77,78], also induces sustained ER stress and activates unfolded protein response (UPR) [79,80]. ER stress triggers dysregulation of several important signaling pathways involved in protein folding [79,80], cell proliferation, survival, and carcinogenesis [77,81]. Furthermore, activation of the ER overload response (EOR) by ER stress results in the activation of NF-κB and p38 signaling pathways as well as elevation of ROS (reactive oxygen species), which leads to oxidative DNA damage and genomic instability [79,80,82]. Additionally, the UPR induced by misfolded pre-S mutants causes oxidative stress, inflammation, tissue damage, cell death, regeneration, and fibrosis. Moreover, truncated pre-S2 mutant protein can act as a trans-activating factor. It can bind directly to the hTERT promoter [69] to activate telomerase expression. Alternatively, it can bind to PKC alpha/beta to trigger the c-Raf-1/MAP2-kinase signal transduction cascade and result in the activation of transcription factors such as AP-1 and NF-κB [68]. Together, HBsAgs can promote cell proliferation, increase cell survival and genetic instability, and cause telomerase reactivation and contribute to the progression from chronic hepatitis to HCC [51,83].

### 2.4. HBV-Induced Inflammation, Liver Injury, and Immunosuppressive Microenvironments Contribute to Hepatocarcinogenesis

Accumulating evidence has demonstrated that inflammation plays an important role in the initiation, promotion, and progression of tumors [84]. HCC represents a classic paradigm of inflammation-linked cancer where various immune and inflammation factors, including T cells [55,85,86] and cytokines, as well as oxidative stress, contribute to hepatocarcinogenesis.

Many studies, including animal models and clinical studies, have shown that HBV infection leads to inflammation and liver injury [55]. It was found that the immune responses in chronic hepatitis B patients are more impaired than those in patients with acute hepatitis B [55]. The immune response in chronic hepatitis B patients is not good enough to clear the virus but can destroy a portion of infected cells in the liver, an organ with unusually high regenerative capacity. Cell death triggers compensatory proliferation. Continuous cell destruction and regeneration leads to liver injury, telomere shortening, telomerase activation and ultimately HCC development.

On the other hand, chronic HBV infection also induced an immunosuppressive microenvironment in the liver. HBV has evolved several strategies to counteract the host immune response for its persistence. HBV infection was found to enhance the immunomodulatory activity of regulatory T (Treg) cells and also increase the population of Treg cells [87]. HBV antigens HBsAg and HBeAg have been shown to impair the functions of NK (natural killer) cells [88]. The immunosuppressive microenvironment aggravates HBV-induced chronic inflammation and helps not only the virus but also tumor cells to escape immunosurveillance [87,89], representing an important mechanism for the progression of chronic hepatitis to HCC.

ROS and oxidative stress can cause damage to cellular DNA, proteins, and lipids, which could further lead to chromosomal mutagenesis and carcinogenesis. Several lines of evidence demonstrated that HBV infection could induce the production of ROS and oxidative stress. It has been found that oxidative stress in the liver of HBV-infected patients was increased, in parallel with increasing viral replication status and disease severity [55,90]. Patients with chronic HBV infection also had significantly higher levels of total oxidative stress and lipid peroxidation compared to patients with inactive HBsAg carrier states and healthy controls [55,91]. Furthermore, DNA damage induced by ROS has been observed in HBV-infected patients. All these data indicate that induction of oxidative stress represents an important mechanism by which HBV infection may drive HCC development. 

## 3. Animal Models of HBV-HCC

Appropriate animal models are valuable to study the pathogenic mechanisms underlying HBV-induced HCC and are indispensable in evaluating the safety and efficacy of new therapeutic strategies before their clinical use. They have the power to recapitulate the complex relationship between the virus, the tumor, and their microenvironment, which are lacking in in vitro systems. In this article, we describe different animal models of HBV- HCC and discuss their advantages and limitations below. For other HCC animal models not directly related to HBV, please refer to review papers [92,93].

### 3.1. Genetically Engineered Mouse Models

HBV is the prototype of the *Hepadnaviridae* family. The viruses in this family display a narrow host range. HBV robustly infects only hepatocytes in humans and higher primates, such as chimpanzees. For a review, please refer to [94]. Given the many advantages of mice in biomedical research in regard to genetic versatility, ease of manipulation, and the wealth of available data, the murine model is the favored one in studying HBV-HCC. Unfortunately, HBV can not infect mice naturally, and thus genetic engineering technologies have been employed to develop various murine models as useful tools in HBV-HCC research. Using transgenic and gene targeting technologies to establish HBV-HCC murine models has the advantage of studying the involvement of specific proteins and signaling pathways in the generation of HCC mechanistically in a scenario that HCC develops spontaneously in a coevolving liver microenvironment with HBV or its gene products [92].

#### 3.1.1. Conventional HBV Transgenic Mouse Model 

HBV complete genome or specific subgenomic fragments have been introduced into the mouse genome via microinjection of HBV DNA into the pronuclei of a fertilized one-cell embryo to establish a variety of HBV-related transgenic mouse lines. These HBV transgenes are usually under the control of either endogenous HBV promoter or exogenous liver-specific constitutive (e.g., albumin promoter) or inducible (e.g., metallothionein promoter) promoters. Although the transgenic mouse models do not entirely mimic the scenario of HBV natural infection in humans, they can support HBV replication and gene expression and thus represent convenient systems to address various aspects of liver diseases caused by the hepatitis B virus, including HCC, in human patients.

In most HBV-related transgenic mouse models, expression of HBV pre-core, core, wild-type small and middle envelope proteins, and even full genome with active viral replication was not necessarily associated with HCC [49,95,96]. Only the transgenic mice with sustained expression of HBx, large HBsAg, and pre-S mutants could develop HCC. These models unravel the importance of these viral oncogenic proteins that, alone or in conjunction with other host cellular genes, promote HCC development via inducing oxidative stress, deregulating host gene expression, and activating oncogenic signal transduction pathways [97,98].

Although not every line of HBx transgenic mice could develop HCC [99,100,101], a high percentage of them developed HCC by 18 months of age [102,103]. They were thus used to study the detailed mechanisms of how chronic HBV infection mediates the occurrence of HCC. It has been well established that HBx could act as a cofactor in carcinogen- and oncogene-, e.g., *c-myc*, induced carcinogenesis in HBx transgenic mice [103]. Moreover, HBx has also been demonstrated to induce malignant transformation of hepatic progenitor cells (HPCs) that contribute to tumorigenesis through activating the oncogenic signaling pathways in HPCs [102]. It was noted that no preneoplastic cirrhosis and inflammation in the liver were observed in the HBx transgenic mice, suggesting that HBx can exert its tumor-promoting activity in the absence of chronic inflammation.

In contrast to the HBx transgenic mouse model, transgenic mice carrying large HBsAg/pre-S mutants do develop obvious inflammation and HCC [104,105]. These mice developed severe and prolonged hepatic injury and progressed to neoplasia within 18 months [106]. Furthermore, like humans, these mice displayed ground glass appearance to hepatocytes, a pathological change associated with overexpression of large HBs/pre-S mutants, and also showed gender disparity, in which male mice develop HCC earlier than female mice. Transgenic mice that overproduced the pre-S1/S2 mutants within hepatocytes can induce oxidative DNA damage in the mouse hepatocytes and cause hepatocarcinogenesis [82]. 

Together, these transgenic mouse models mimic many pathological events that occur before the development of HCC in humans with chronic HBV infection. Therefore, they could serve as good models to investigate the different roles of HBsAg and HBx and underlying mechanisms in early events of HBV-related hepatocarcinogenesis, as well as to test new therapeutic strategies for HCC.

#### 3.1.2. Viral Vector-Mediated Transgenic Mouse Model

Although HBV transgenic mice are very valuable animal models, the acquisition of transgenic mice is very time-consuming, laborious, expensive, and needs extensive infrastructures and complicated technologies. Furthermore, the immune system of the mouse recognizes the protein product of the transgene as a “self-antigen” and develops a tolerance to it. This makes the study of immune-related mechanisms or therapeutic approaches difficult in these models. Alternative models that are less laborious, time-consuming, have fewer technical requirements, and allow more flexibility in transgene combinations can be considered to substitute for the transgenic mouse models in some studies.

Recombinant adeno-associated virus (rAAV) has been recognized as a promising tool for delivering genes in vivo [107,108,109]. AAV vectors can stably express gene products from unintegrated episomes without modifying the host genome. In addition, AAV vectors are particularly useful for long-term experiments in vivo because of characteristics such as the lack of pathogenicity, low immune responses, and a wide range of cell tropism provided by different serotypes. By using hepatotropic rAAV carrying HBV DNA (AAV/HBV), persistent HBV DNA and protein expression can be achieved in the liver of mice, mimicking chronic infection in humans [110,111]. Importantly, similar to the clinical HBV carriers, the mice infected with rAAV/HBV remained seronegative for HBsAg antibodies [111] and developed histopathological alterations which are consistent with fibrosis [112] and HCC [110]. Co-delivery of oncogenes or shRNAs against cellular RNAs and HBV DNA with rAAV can further dissect the roles of specific genes or signal transduction pathways in HBV-HCC. Accordingly, rAAV/HBV represents a useful alternative model to study the pathogenic mechanisms of HBV-associated HCC and the development of HCC therapeutic drugs.

Although rAAV is defective in replication and only allows one-cycle infection, it still has the possibility to infect humans. Therefore, potential biohazards associated with the delivery of infectious agents (such as HBV) or genes encoding potentially toxic or tumorigenic gene products with rAAV need to be carefully evaluated when using rAAV-mediated mouse models. The trans-splicing-mediated AAV gene delivery technique can be employed to minimize the risks of biohazards. The gene (cDNA) encoding the potentially hazardous product is split into two independent AAV transfer plasmids and flanked with intron donor and acceptor signals, respectively. The functional product can only be restored when the two rAAV vectors are co-administered into the same cells, undergo a concatemerization process through intermolecular recombination as well as transcription and splicing across the inverted terminal repeat junction in the reconstituted genome in the same cells [113,114]. This technique also allows the expression of transgenes exceeding the packaging limitation of individual rAAV vectors, expanding the utility of rAAV-mediated mouse models.

#### 3.1.3. Hydrodynamics-Based Transfection (HDT) and Genetic Modification Systems

Hydrodynamics-based transfection (HDT) is a non-viral method to deliver genetic materials into the hepatocytes of mice through intravenous injections of plasmids encoding the target gene of interest (e.g., HBV) in large (8~10% of body weight) volumes of saline over a short period of time (5–7 s) [115,116]. This technique provides a quick and convenient method to deliver genes of interest alone or in combination simultaneously to the hepatocytes of the mouse. It has been widely used to study the mechanisms of a variety of liver diseases. By injecting HBV replication-competent constructs, acute or persistent HBV replication can be established in immune-competent mice depending on the HBV constructs and mouse strains [116,117,118,119,120]. Liver fibrosis could be observed in mice receiving recombinant HBV covalently closed circular DNA [120]. By combining HDT technology with different genetic modification systems, such as HBV transgenic mouse models, CRISPR/Cas9 genetic editing [90], and the Sleeping Beauty (SB) transposon system [91], HBV-HCC models have been established for mechanistic studies.

One major advantage of using HDT to establish HCC mouse models over conventional transgenic mouse technology is the flexibility in terms of transgenes and strains of the recipient mice, thereby saving time and cost. Employing the SB transposon system or CRISPR/Cas9 genetic editing with HDT, long-term expression of different combinations of oncogenes (e.g., *NrasV12*, *myr-AKT*, *c-Myc*, *ΔN90*-*β-catenin*, and HBx, etc.) or shRNAs against tumor suppressor genes (e.g., *shp53*) can be achieved to induce the transformation of hepatocytes [121,122]. For instance, delivery of an HBx gene constructed in the SB transposon system into the livers of fumarylacetoacetate hydrolase (*Fah*) mutant mice via the HDT method activates the expression of *β-catenin*, one of the most frequently activated genes in human HCCs [123], and induces hepatic inflammation. The co-administration of *shp53* with HBx further accelerates the formation of liver hyperplasia [122]. In another study, CRISPR/Cas9 system with *p53* and *Pten* dual sgRNA expressing cassette was delivered to the liver of transgenic mice carrying the HBV large envelope polypeptide [89]. Disruption of *p53* and *Pten* genes in this HBV transgenic mouse model accelerated tumor formation from 12 to 20 months to as early as 4 months post-HDT. Accordingly, genetic features found in patients with liver cancers can be recapitulated in mice models with shortened latency and increased tumor formation by these approaches. These models are thus very useful for dissecting the roles of different genetic alterations and in vivo functional validation of specific viral or cellular genes in hepatocarcinogenesis [124,125,126] as well as for testing molecularly targeted anti-HCC therapies.

### 3.2. Hepadnavirus Natural Infection-Induced HCC Model 

HBV can not infect mice and other conventional laboratory animals. Although mice genetically modified to express HBV proteins or support viral replication can be used as convenient and useful tools in many aspects of HBV-HCC research as described above, none of these mouse models can totally mimic the human situation in which liver tumors arise in the background of chronic HBV infection. Therefore, other hepadnaviruses-infected animals that can more faithfully reflect the situation of human HBV-HCC have been explored both for basic studies of tumor biology and for experimental therapeutic purposes. The most commonly used surrogate models for HBV infection are HBV-infected chimpanzees, HBV-infected tree shrews, woodchuck hepatitis virus (WHV)-infected woodchucks, ground squirrel hepatitis virus (GSHV)-infected ground squirrels, and duck hepatitis B virus (DHBV)-infected Pekin ducks. Chronic hepatitis can be developed in these animal models, however, in contrast to the human situation, liver cirrhosis has been observed only in few cases [127,128,129], and only persistent WHV infection is consistently associated with a high incidence (almost 100%) of HCC. Development of HCC has also been reported in less than 10% of GSHV-infected ground squirrels [130] and 33% of HBV-infected tree shrews [131].

#### 3.2.1. Tree Shrew Model of HBV-HCC

Northern tree shrews, *Tupaia belangeri*, are small diurnal animals, genetically more closely related to primates than to rodents. Tupaias are the only non-primates susceptible to the hepatitis B virus. Tree shrews have a greater propensity to progress to chronic hepatitis when infected as neonates and exhibit similar pathological changes in the liver as those in humans. As in humans, the oncogene *p53* was mutated in the tree shrew model for HCC [42]. In addition, the upregulation of the p21 protein and the activation of oncogene *Ras* were observed in the early stage of hepatocarcinogenesis [132]. Levels of CuZn-superoxide dismutase (SOD1) and glutathione S-transferase A1 (GSTA1) were significantly decreased in both tree shrew and human HCC tissues, and the downregulation of these two proteins may cause persistent oxidative damage-inducing hepatocarcinogenesis [133]. These characteristics make the tree shrew an ideal model for HBV-HCC research. However, a number of limitations have decreased their usefulness and ease of use, including the overall low viral titer and persistent rate, the long period of time required to develop the tumors (5–6 years post-infection), and the lack of many research tools and reagents for this species. 

#### 3.2.2. Woodchuck HCC Model: A Surrogate Model Based on HBV-Related Hepadnavirus

The Eastern woodchuck (*Marmota monax*) can be naturally and experimentally infected with woodchuck hepatitis virus (WHV), a hepadnavirus closely related to the HBV. Similar to HBV-HCC in humans, HCC spontaneously develops in woodchucks in the background of chronic WHV infection. The rate of chronic WHV infection following neonatal inoculation is 60% or higher, and virtually all woodchucks chronically infected with WHV develop liver tumors within the first 2–4 years after infection at birth with an approximate 6-month life expectancy after that [134,135]. Similar to HBV-infected patients, liver inflammation, injury, and repair processes also occur in the WHV-infected woodchucks. Therefore, the time required for the progression from chronic infection to HCC in humans can be greatly shortened, and the entire process of hepatocarcinogenesis can now be monitored and studied within a reasonable time frame in this model. 

The tumors in woodchucks are comparable in size, morphology, pathological changes, and molecular gene signature to those of HBV-HCCs [128]. Like HBV, WHV DNA can also integrate into the host genome; however, important differences should be noted. WHV DNA is integrated into hepatocyte DNA within or nearly contiguous to a specific locus associated with overexpression of *c-myc* and *N-myc* genes with high frequency [136,137], while the insertion of activated dominant oncogene is not a typical mechanism for HBV carcinogenesis. Moreover, WHV DNA integration causes genetic remodeling, such as *N-MYC2* rearrangements, that provides a proliferative stimulus or growth advantage for transformed hepatocytes. A study using transcriptomic analysis has revealed that WHV-induced HCC is positively correlated with the S2 subclass of human HCC, which is associated with MYC activation, alpha-fetoprotein (AFP), and epithelial cell adhesion molecule (EpCAM) expression [5]. Furthermore, like HBx, the X protein of WHV also acts as a multifunctional transactivator and is pivotal for WHV infection as well as the malignant transformation of hepatocytes [138].

Apparently, the oncogenesis of WHV-related HCC recapitulates that of HBV-HCC in many aspects. These characteristics make the WHV-infected woodchuck a more clinically relevant model for human HCC and can be used to address certain critical questions that cannot be completely explored in humans. In addition, the larger size of the woodchuck (about the size of a human baby) allows the direct application of clinical facilities and human-size products to this model, making it a better preclinical model for testing novel advances in imaging and therapeutic techniques. The woodchuck has proven to be suitable for performing CT, MRI, liver embolization, and arterial catheterization to evaluate various intra-arterial therapies [139,140]. Results from the woodchuck model can be more readily translated to humans than other small animal HCC models. For a comprehensive overview, please refer to the review [109].

Lack of research tools and woodchuck-specific reagents significantly impedes the use of this model to explore the molecular mechanisms of hepadnavirus-related hepatocarcinogenesis in the woodchuck model. Another potential limitation is that the molecular mechanisms underlying HCC formation in woodchuck HCC may not represent all types of human HCCs. Recently, the whole genome sequence of the Eastern woodchuck has been established [114]. Advances in the genome information of this species will facilitate the use of the woodchuck model for further mechanistic studies in chronic hepatitis B and hepatocellular carcinoma. 

### 3.3. Human Liver-Chimeric Mice Model

Chimeric mice with humanized liver contain repopulated human hepatocytes in the majority of the liver. To establish engraftment, suitable recipient mice must be immunodeficient to prevent xenograft rejection, and they also exhibit endogenous liver damage to allow expansion of the transplanted hepatocytes. For more detailed information about the establishment of human liver-chimeric mice, please refer to the review [94,141,142].

In contrast to genetically engineered mouse models, the human liver-chimeric mice are susceptible to HBV infection and are capable of forming HBV cccDNA as well as supporting HBV replication. Considering chronic hepatitis B is a host-specific immune-mediated liver disease, one of the shortcomings of human liver-chimeric models is their highly immunodeficient background. Thus, in order to enable the study of HBV-related immunopathogenesis, the dual chimeric mouse models with simultaneous engraftment of both hepatocytes and human hematopoietic stem cells (HSCs) are gradually developed. However, the establishment of dual chimeric mice, particularly those with efficient reconstitution of the human immune system, may be hampered by technical difficulties.

Importantly, after HBV infection, the dually engrafted mice sustained high viremia, exhibited specific immune and inflammatory responses, and showed the progression from chronic hepatitis to liver cirrhosis, but HCC development was not observed [142,143]. Because HCC takes a long period of time to form, it may be difficult for dual chimeric mice to recapitulate such HBV-associated liver pathology. Nevertheless, this in vivo model may still be helpful in terms of elucidating oncogenic pathways involving early phases of HCC initiation and progression. Engraftment of human liver cells expressing oncogenes or shRNAs against cellular RNAs and HBV DNA may help dissect the roles of specific genes or signaling pathways in HBV infection as well as HBV-related hepatocarcinogenesis. Accordingly, the dual chimeric mouse represents an ideal small animal model for HBV-related research and may be acutely required in the future.

The versatility of these HBV-HCC animal models is expected to broaden our knowledge of the genetic alterations underlying HBV-induced hepatocarcinogenesis, allowing the study of malignant liver lesions and the evaluation of novel therapeutic strategies and potential biomarkers. The comparison of different HBV-HCC animal models with methodologies is summarized in Table 1.

## 4. Relationship between the MTA1 and HBV-HCC

The lack of sensitive biomarkers for timely diagnosis and effective therapeutics for advanced tumors are two major reasons for the poor outcome of HCC, showing an imperative need to identify potential markers and therapeutic targets of HCC. The most well-studied HCC biomarkers are the alpha-fetoprotein (AFP), its isoform AFP-L3, and des-γ-carboxy prothrombin (DCP). However, the current commonly used markers, such as AFP, cannot effectively predict tumor recurrence. In an effort to identify novel markers for the detection of HCC recurrence, many other molecules have been explored, including glypican 3 (GPC3), cytokeratin 19 (CK19), and glutamine synthase (GS), etc. For more detailed information, please refer to the review of Zacharakis et al. [144].

The metastatic tumor antigen 1 (MTA1) is closely correlated to poor prognosis and frequent postoperative recurrence in patients with HCC, particularly in those with HBV-HCC [145,146,147]. Thus, MTA1 has the potential to serve as a prognostic marker and therapeutic target for HBV-HCC. Most of the current understanding of the MTA1 functions in HCC and other cancer types is derived from in vitro studies. However, the need for exploration of the clinicopathological significance and potential clinical applications of MTA1 cannot be satisfied completely with in vitro experiments. Suitable animal models are essential to advance our understanding of the underlying molecular and pathophysiological mechanisms with respect to MTA1 in HBV-HCC development and to develop new therapeutic strategies.

In our previous work, we demonstrated the translational value of the woodchuck model and characterized the molecular functions of MTA1 in hepadnavirus-induced HCC through an interaction with the viral X protein [33]. Herein, we used several animal models, including the woodchuck model, to study the roles of MTA1 in HBV-HCC as an example for investigating the specific questions which cannot be completely explored in humans. We also briefly summarized the current knowledge of MTA1 with respect to HBV-HCC development and discussed the relevant experimental results and the caveats surrounding the use of these animal models.

### 4.1. Structure and Molecular Function of MTA1: An Overview

MTA1 protein is a major component of the nucleosome remodeling and histone deacetylase (NuRD) complex [146,148]. It consists of 715 amino acids and contains several crucial structural domains and motifs [149] (Figure 3). The first domain in the N-terminal portion of MTA1 is the Bromo-adjacent homology domain (BAH) which has a role in protein-protein interactions. Next to the BAH domain, there is an ELM2 (Egl-27 and MTA1 homology) domain and a SANT (SWI, ADA2, N-CoR, TFIIIB-B) domain, which are charged with recruiting the chromatin remodeling enzymes for transcriptional regulation [150,151]. Following the SANT domain is a GATA-type zinc finger domain (ZnF) which plays a role in the direct interaction of MTA1 with transcription factors and transcriptional co-regulators [152]. In addition, another possible src-homology (SH)-binding motif was observed; five SPXX motifs were found close to the carboxyl terminus of MTA1 protein and three putative nuclear localization signal sequences located at the C-terminal end of MTA1 protein [153,154]. For discussion of the identification and characterization of the MTA1 gene and its encoded proteins, please refer to the reviews contributed by Toh and Nicolson [146,155]. 

MTA1 has been considered a multifaceted mater coregulator that can either act as coactivator or corepressor to modulate the expression of target genes and/or the activity of its interacting proteins [146,148,156]. In this case, MTA1 can control a spectrum of cancer-promoting processes, including transformation, proliferation, invasion, and metastasis, as well as angiogenesis [148,157]. The cancer-relevant functions of MTA1 may be the result of the following mechanisms: post-translational modifications of MTA1 in response to upstream signals; interaction with multiple molecules with established roles in cancer-relevant processes; and modulation of expression of a variety of target genes by its dual transcriptional coregulatory activity. The above-mentioned mechanisms have been extensively observed in many cancer types, including HCC. For more detailed information, please refer to the reviews [148,157].

We previously reported an exon 4-excluded form of MTA1, MTA1dE4, as shown in Figure 3. According to the gene structure, the difference between MTA1 and MTA1dE4 is in the BAH domain, which plays an important role in regulatory protein-protein interactions, nucleosome binding, and recognition of methylated histones. Therefore, MTA1 and MTA1dE4 may interact with different regulatory proteins and differ in the functions associated with BAH activities. Our previous study showed that MTA1dE4 exhibits a greater ability in promoting the migration and invasion of HCC cells than that of the full-length MTA1. However, the potential mechanisms remain unclear and need further investigation. The possible functions of MTA1dE4 and its differences from the full-length MTA1 are summarized in Table 2. 

### 4.2. Regulation of MTA1 Expression in HBV-HCC

HBV-induced hepatocarcinogenesis is a multistep process associated with changes in host gene expression, some of which have transforming and oncogenic potential. The HBV regulatory X protein (HBx) has been demonstrated to be involved in numerous mechanisms of oncogenesis by its interaction with a variety of host proteins [158].

MTA1 is a cancer-promoting molecule and has attracted wide attention because of its interaction with HBx contributing to HCC development. Our previous work and other studies have demonstrated that MTA1 is overexpressed in HBV-HCCs [145,146,147,159,160], and the hepadnavirus X protein can induce the expression of MTA1 through activating the NF-κB signaling pathway in HCC cells. In addition, HBx can also transactivate the expression of MTA1 by c-MYC and tumor growth factor-β (TGF-β). Similar to NF-κB, c-MYC can also bind directly to the *MTA1* promoter to increase the transcription of *MTA1* [161,162]. In addition, the CDP (CCAAT displacement protein) can recruit tumor growth factor-β (TGF-β) to the *MTA1* promoter and upregulate its expression [163,164]. 

Few studies have reported the potential role of MTA1 in HBx-mediated hepatocarcinogenesis [59,165]. In this process, MTA1 has been demonstrated to play an integral role in the HBx protein stimulation of NF-κB signaling [59] and stabilization of hypoxia-inducible factor-1 alpha (HIF-1α) [166]. These findings provide further evidence that MTA1 is required for HBx transactivation function and make it an important molecule of HBx-mediated hepatic disease progression.

In addition to the HBx protein, many factors, such as oncogenes, hypoxia, inflammation, and growth factors, etc. [148], are capable of inducing MTA1 expression. For more detailed information about other upstream modifiers of MTA1 protein and its downstream effectors and targets, please refer to the review [148].

### 4.3. MTA1 Is Overexpressed in HBV-HCC

Hamstsu et al. were the first to report the correlation between the malignancy of HCC and the overexpression of MTA1 in liver cancer tissues [159]. In addition, they reported that MTA1 could be used to predict the post-operative survival rate of patients with HCC. Other studies further showed that the median survival rate in patients with high MTA1-expressed tumors is significantly lower than that in patients with low MTA1-expressed tumors [160]. Moreover, MTA1 overexpression in patients with HBV-HCC was significantly associated with several clinical factors, including younger age, female gender, higher serum alpha-fetoprotein level, and Child-Turcotte-Pugh class A [160]. In our previous work, as well as in other studies, MTA1 has been found to be associated with more aggressive clinicopathological features in patients with HCC, especially in those with HBV-HCC, and thus could be a marker for postoperative early recurrence and poor prognosis in these patients [145,146,147]. On the whole, these clinical observations indicated that the overexpression of MTA1 is strongly associated with hepatocarcinogenesis and malignant features, including invasion and metastasis.

### 4.4. Therapeutic Significance of MTA1

The prognosis of a patient with advanced HCC is poor partially due to therapeutic resistance to traditional chemotherapeutic agents (e.g., Sorafenib). Therapeutic resistance of cancer is an important factor in contributing to the invasiveness and metastasis of cancer cells [167,168], which relies on escaping apoptosis and increasing drug efflux [169]. In recent years, several studies have focused on MTA1 as an important mediator of cancer therapeutic resistance [170,171,172,173,174]. Despite the fact that the role of MTA1 in therapeutic resistance has not been studied and discussed on HCC, these studies provide information about emerging functions and mechanisms of MTA1 in aggressive cancers.

Moreover, MTA1 is gradually perceived to be a new target candidate for cancer drug treatment. Recently, MTA1-targeted chemopreventive agents for cancer therapy have been continuously developed and studied [175,176,177]. For instance, natural compound resveratrol and its analog pterostilbene have been recognized as the potent inhibitors for MTA1 and MTA1-guided signaling [178,179,180]. These compounds appear to regulate the PTEN/AKT and P53 signaling pathways through the inhibition of the MTA1/HDAC unit of the NuRD complex in cancer cells and finally inhibit tumor growth, progression, and metastasis of cancers. A clinical trial evaluating the therapeutic efficacy of pterostilbene, an MTA1 inhibitor, for the treatment of prostate cancer has been designed and is ongoing [Levenson, A.S. (2018) MTA1-targeted strategies for prostate cancer management. Identification No. 9511118. Retrieved from https://reporter.nih.gov/project-details/9511118#details (accessed on 22 August 2021)]. Although this project is not being conducted for HCC, it still offers clinical proof for pterostilbene as a promising natural agent for MTA1-targeted chemopreventive and therapeutic strategies to curb cancer. Together, these findings highlight the potential of using MTA1 as a promising therapeutic target and facilitating MTA1-targeted strategies, as well as developing combinatorial therapeutic approaches for cancer treatment in the future.

### 4.5. Application of Hepadnavirus-Induced Woodchuck HCC Model for Studying the Biological Functions and Clinical Significance of MTA1 in HBV-HCC

Our previous study [33] provided several lines of evidence to demonstrate that an HBV natural infection model, the woodchuck hepatitis virus (WHV)-infected woodchuck, is an appropriate model to study the role of MTA1 in hepadnavirus-induced HCC by characterizing the molecular function(s) of woodchuck MTA1. Similar to human MTA1, woodchuck *MTA1* (wk-*MTA1*) was overexpressed in WHV-induced HCC of the woodchuck and played an indispensable role in the hepadnavirus X protein-mediated NF-κB activation and cell migration. In this study, we were the first to reveal that the expression of a major spliced variant of wk-*MTA1*, *MTA1dE4* (exon 4-excluded form of *MTA1*), but not the total wk-*MTA1*, is associated with more malignant characteristics of woodchuck HCC, including multiple tumors and a larger size.

Using the woodchuck model, we studied the complex temporal relationships between *MTA1*/*MTA1dE4* expression and the progression of HBV-related liver diseases, including chronic hepatitis and HBV-HCC. For instance, we monitored the dynamic changes of target gene expression (e.g., *MTA1* and *MTA1dE4*) in the liver of a woodchuck after experimental WHV infection within a reasonable time period (Figure 4). Moreover, we assessed the expression of the full-length *MTA1* (*MTA1-FL*) and the spliced variant *MTA1dE4* in both tumor and non-tumor tissues, all of which were obtained by serial liver biopsy at different time points, and analyzed the correlation between them and the size of the tumor, as well as the result of serological assay (Figure 5). It can simultaneously implement various technologies to monitor the tumor take rate and growth rate in woodchucks as well as to correlate histopathological results at various time points with serum and biomarkers, which cannot be longitudinally determined in patients. By taking this advantage, it can offer a pre-clinical model for testing the promising MTA1-targeted chemopreventive agents and therapeutic strategies to curb HBV-HCC.

### 4.6. Translational Researches Based on the Woodchuck Model

For decades, researchers have been exploring more sensitive biomarkers and developing more effective therapeutic strategies for HCC patients with a high risk of tumor recurrence. However, a great number of patients with HBV-HCC still suffer tumor recurrence [181,182]. Our woodchuck HCC study demonstrated that *MTA1dE4*, a major spliced variant of *MTA1*, may represent a more sensitive marker than total *MTA1* in WHV-induced HCC [32]. We thus analyzed the relationships between clinical characteristics of patients with HBV-HCC and expression of total *MTA1* and *MTA1dE4*. We also explored the clinical impact of MTA1 and one of its major spliced variants, *MTA1dE4*, on postoperative recurrence in patients with HBV-HCC in Taiwan through a 4-year retrospective cohort study [32]. For the first time, we revealed the clinical significance of *MTA1dE4*. It overexpressed in a higher percentage than total *MTA1* in HBV-HCC patients and can serve as a more sensitive marker to predict the onset of early recurrence of HCC, especially in patients with low AFP. 

Therefore, we successfully translated the results of basic science research into human research and showed the potential clinical application of MTA1 in HBV-induced hepatocarcinogenesis. This might represent a demonstration that new findings yielded by the woodchuck HCC model could be extrapolated and applied to HBV-HCC.

### 4.7. Potential Issues on Application of HDT-Based Murine Models for Oncogenic Collaboration Studies in HBV-HCC

The flexibility of the HDT technique in terms of transgenes and strains of the recipient mice renders it the ideal approach to determine the in vivo oncogenic potential of the MTA1 and MTA1dE4. In this case, we established an HDT-based HCC mouse model in C57BL/6JNarl mice with co-administration of *NrasV12*/*MYC* oncogenes either with *MTA1* or *MTA1dE4* to assess their additive effect in terms of promoting tumorigenesis. Consistent with previous studies [121,183], we found that hydrodynamic injection of *NrasV12*/*MYC* oncogenes induced the development of nodular and diffuse HCC within 8 weeks in some mice (Figure 6A). However, we observed that the variability of tumor burdens (including size and number) among mice in the same group is too large to investigate the additive effect of MTA1 or MTA1dE4 on accelerating tumorigenesis. We suggested that the variability in tumor burdens was mainly due to different expression levels of oncogenes, which were caused by the close-to-random insertions of transposons. 

Furthermore, we also speculated that the genetic background, in addition to the close-to-random insertion effect, could be a possible factor affecting the rate of HCC formation in distinct mouse strains. We stably co-expressed *NrasV12* and *c-MYC* oncogenes in the livers of different mouse strains (e.g., CBA/CaJNarl, C57BL/6Jnarl, and C3H/HeNCrNarl strains) with the same approach and observed a dramatic divergence in both tumor latency and tumor burdens among these mouse strains. (Figure 6B). The possible explanation for this phenomenon awaits further investigation.

## 5. Concluding Remarks

Multiple mechanisms are involved in the progression of chronic liver disease to HCC formation in patients with chronic HBV infection. Animal models of HBV-HCC have contributed to our understanding of hepatocarcinogenesis and cancer progression. From the animal models described in this review, several features are highlighted. First, tumor development in models is much slower by the single HBV gene transfer than by the delivery of the HBV gene in conjunction with other genes (e.g., oncogenes and shRNAs) or chemical exposure. Secondly, genetically engineered mouse models do not develop liver fibrosis and cirrhosis, suggesting these symptoms observed in HCC patients could be contributed by factors other than HBV itself. Finally, these models cannot encompass the full clinical picture in humans and may result in a skewed model. To overcome a potential bias in terms of strain-dependent effects, research should be replicated across different models, strains, and genders and must be as close as possible to mirror the conditions in humans. 

In this review, we used MTA1 as a molecular target model to demonstrate the investigation of HBV-related hepatocarcinogenesis and the applications of animal models for this purpose. We shed light on the fact that MTA1 can be clinically useful for the prediction of the progression of HBV-HCC. Thus, evaluating the expression levels of MTA1 in individual cases may provide clinicians with important clues for developing prognosis and possible therapeutic strategies.

## Figures and Tables

**Figure 1 ijms-22-09380-f001:**
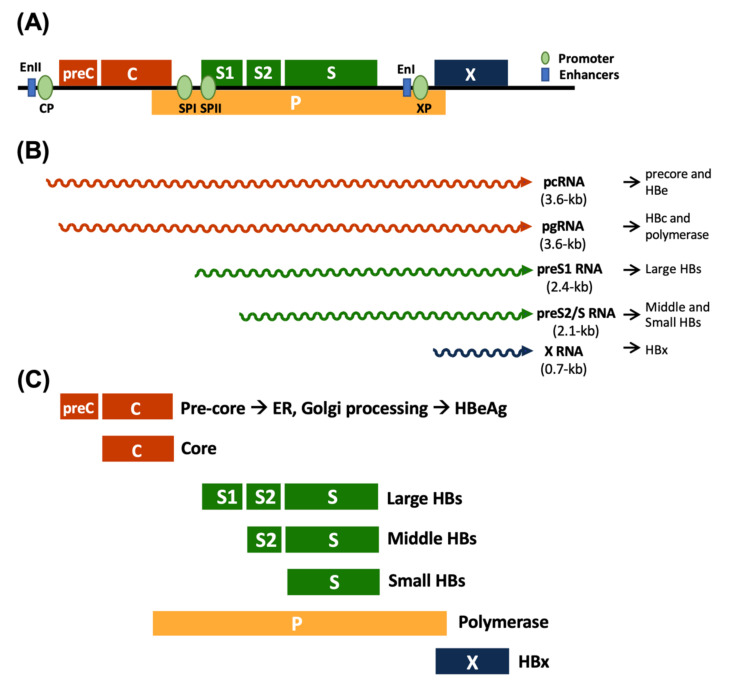
HBV genome, genes, transcripts, and proteins. (**A**) HBV genome organization. (**B**) Major overlapping ORFs for 3.6, 2.4, 2.1, and 0.7 kb HBV RNAs. (**C**) The ORFs encoding the HBV surface proteins (green), the multifunctional viral polymerase (yellow), the secretory HBeAg and capsid-forming core proteins (orange), and the regulatory HBx protein (dark blue) are shown. pcRNA, pre-core RNA; pgRNA, pregenomic RNA.

**Figure 2 ijms-22-09380-f002:**
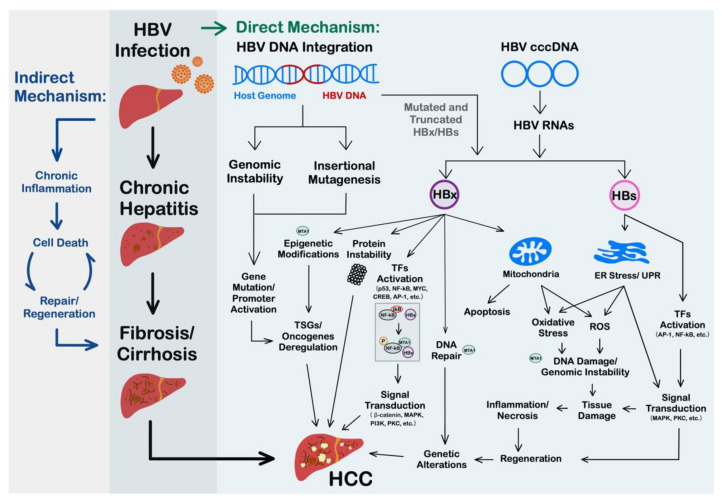
Mechanisms of HBV-related hepatocarcinogenesis. The direct mechanisms of HBV-induced HCC, including HBV DNA integration and persistent expression of viral proteins such as HBx and HBs, can activate cancer-related gene expression and induce genetic instability and oxidative stress. The oncogenic process is promoted by a variety of pathways and mediated by both the host and viral signaling events. In the content of indirect mechanism, the chronic inflammation triggered by host immune responses contribute to ceaseless hepatocyte cell death–regeneration and provides a favorable ground for the emergence of genetic and epigenetic alterations leading to hepatocyte transformation, as well as HCC progression. Furthermore, the potential roles of MTA1 in HBV-related hepatocarcinogenesis are also presented. TSGs, tumor suppressor genes; TFs, transcriptional factors; ER, endoplasmic reticulum; ROS, reactive oxygen species.

**Figure 3 ijms-22-09380-f003:**
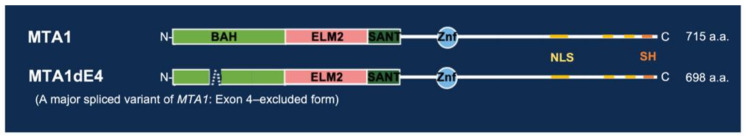
Schematic representation of structural domains of MTA1 and its exon 4-excluded form, MTA1dE4. NLS, nuclear localization signal; SH, src-homology binding motif.

**Figure 4 ijms-22-09380-f004:**
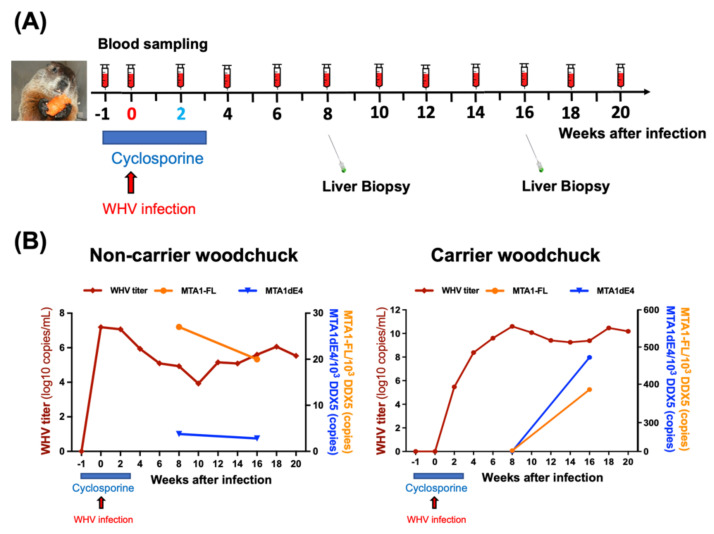
Both MTA1-FL and MTA1dE4 were significantly increased in the liver of woodchuck after WHV infection. (**A**) A schematic illustration of the experimental procedure. To increase the possibility of chronic infection, the woodchucks were treated with cyclosporin to inhibit immune response a week before WHV infection and continued for three more weeks after infection. Blood and liver samples were serially collected at various time points. (**B**) Dynamic change of WHV DNA titer, *MTA1-FL* mRNA, and *MTA1dE4* mRNA in both non-carrier (left panel) and carrier (right panel) woodchucks. Blood samples of the woodchucks were taken periodically, and the presence of WHV was confirmed by using RT-qPCR with the specific primers. As shown in the right panel, a woodchuck has become a carrier after WHV infection and co-administration of cyclosporin. Expression of MTA1-FL and MTA1dE4 mRNA in both non-carrier and carrier woodchucks were examined by RT-qPCR.

**Figure 5 ijms-22-09380-f005:**
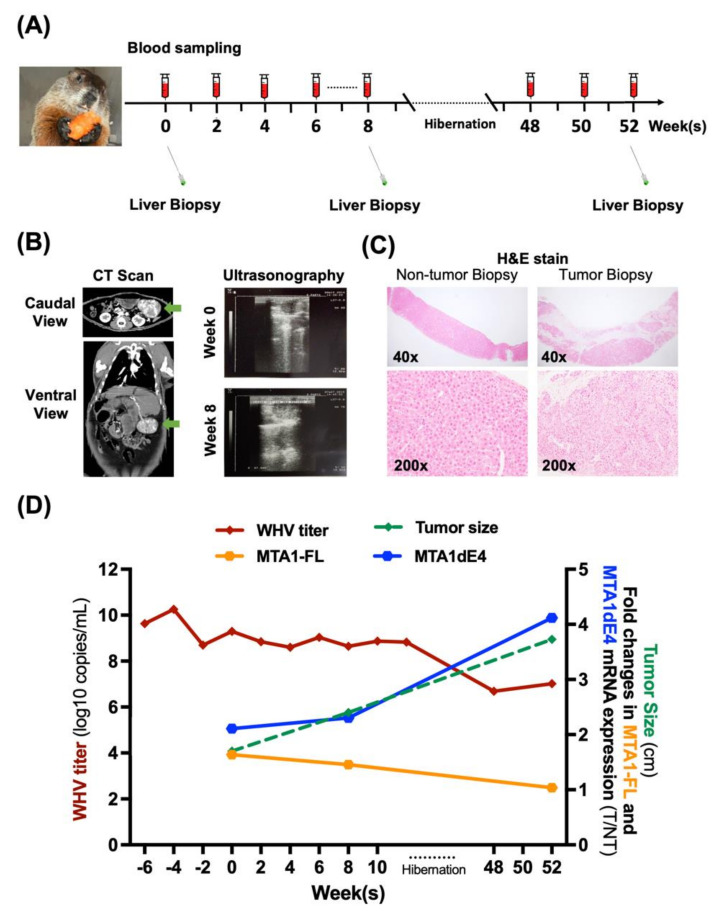
An example for using the woodchuck HCC model to study the complex temporal relationships between *MTA1*/*MTA1dE4* expression and the progression of chronic liver disease. (**A**) A schematic depicting the experimental procedure of conducting longitudinal monitoring of tumor growth and serial collection of serum and liver samples on a woodchuck HCC model. (**B**) Localization of liver tumors by computed tomography (left) and ultrasonography (right). The liver of a woodchuck with one large HCC is indicated by the green arrow. (**C**) Liver biopsies of non-tumor and tumor were stained by the hematoxylin and eosin (H&E) stain and are shown at 40× and 200× magnifications. (**D**) Dynamic change of WHV DNA titer, *MTA1-FL* mRNA, and *MTA1dE4* mRNA, as well as tumor size in a chronically WHV-infected and HCC–bearing woodchuck.

**Figure 6 ijms-22-09380-f006:**
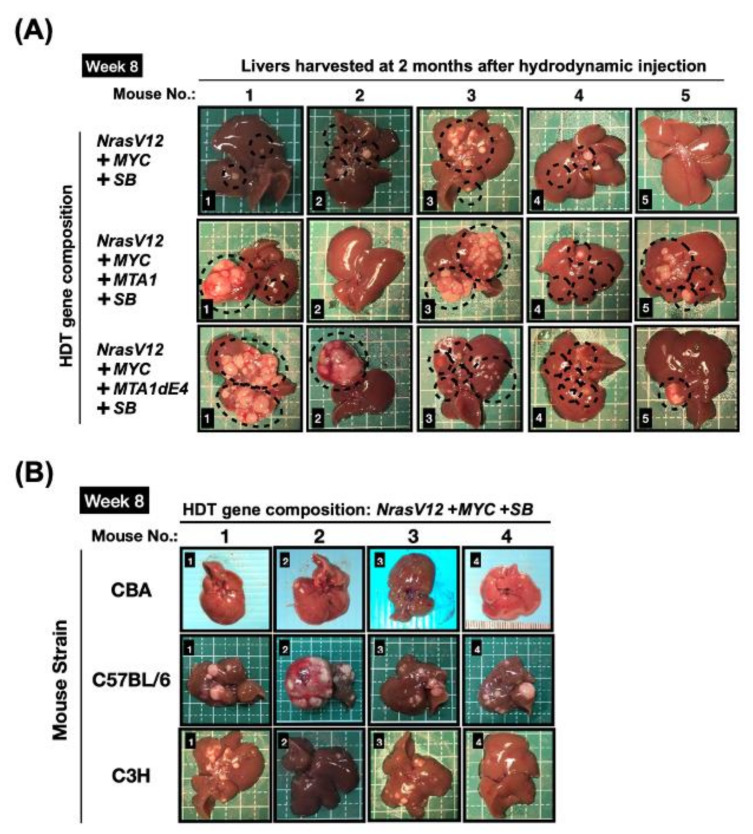
Application of the HDT-SB transposon methodology to investigate the collaboration effects of oncogenes and MTA1 as well as MTA1dE4 in terms of liver tumor induction. (**A**) Selected transposons were hydrodynamically delivered to the liver of C57BL/6JNarl mice together with plasmids encoding SB transposase. Gross morphology of representative livers harvested at 8 weeks post-HDT (*n* = 5 per group). The position of the tumor is indicated by a white dashed circle. (**B**) The gross morphology of representative livers expressing the *NrasG12V* and *MYC* transposons as well as the SB transposase in the context of different mouse strains (*n* = 4 per mouse strain).

**Table 1 ijms-22-09380-t001:** Comparison of HBV-HCC Animal Models.

Method	HCC Development	Advantages	Disadvantages	Features
Conventional HBV transgenic mice	HBV Tg: not observed	No genetic variation in established mouse line	Costly, laborious, and time-consuming	Inbred
	HBx Tg: within 18 mo., but not every line developed HCC		Technically challenging	HBx Tg: no inflammation
	LHBs/pre-S mut. Tg: within 18 mo.		Resource-demanding	LHBs/pre-S mut. Tg: inflammation
Viral vector-mediated transgenic mice	Varied (depending on the properties of the transgene), e.g., rAAV/HBV Tg: 12–16 mo.	Easy and simple procedure	Potential biohazards	Inbred rAAV/HBV Tg: occasionally displayed mild to moderate liver fibrosis
Hydrodynamics-based transfection (HDT)	Sleeping Beauty: varied (depending on the properties of the transgenes)	Easy and simple procedure Flexibility in gene delivery Combination of various genetic modifications possible	Random integration of transgene Genetic variation within a model	Inbred Different HCC latency periods were observed among distinct mouse strains
	CRISPR/Cas9: varied	Easy and simple procedure Combination of various genetic modifications possible	Possible genetic variation (e.g., off-target effect)	Inbred Modification of endogenous genes (e.g., tumor suppressor)
Tree shrew HCC model	Within 5–6 years after an HBV infection in the background of chronic viral infection	Only non-primates susceptible to HBV A greater propensity of progressing to chronic hepatitis when infected as neonates	Overall low viral titer and persistent rate Long period of time to develop the tumors Resource-demanding Lack of many research tools and reagents	Outbred Similar pathological changes in the liver as those in humans
Woodchuck HCC Model	Within 1–4 years after a WHV infection in the background of chronic viral infection	Naturally or experimentally infected by WHV Longitudinal monitoring possible (e.g., CT, MRI, US, etc.) Serial sample collection available (e.g., serum, liver biopsy, etc.)	Costly Research tools and reagents are limited	Outbred Comparable to human HBV-HCCs in tumor size, morphology, pathological changes, and molecular gene signature
Humanized mice (human hepatocytes and/or immune cells engraftment)	HCC not found HBV-associated liver pathology was observed (e.g., chronic hepatitis, fibrosis, and cirrhosis)	Supporting HBV infection and replication Capable of engrafting clinical specimens and genetically modified liver cells Functional human immune system	High costs Technically challenging Resource-demanding	Inbred Similar to CHB patients in respect of high viremia, inflammatory and immune responses

Abbreviations: Tg., transgenic mice; LHBs, large HBx antigen; mut., mutant; mo., month; CT, computed tomography; MRI, magnetic resonance imaging; US, ultrasonography; CHB, chronic hepatitis B.

**Table 2 ijms-22-09380-t002:** Structural, molecular, and clinical insights into MTA1 and MTA1dE4.

	Properties		MTA1	MTA1dE4
Structural insight	Protein size (amino acids)		715	698
	Structure of BAH domain		Long disordered region within the BAH domain (Without regular secondary structure)	Deletion at the disordered region may cause conformational changes of BAH domain of MTA1dE4 *
Molecular insight	Protein-Protein interaction		With common and unique interacting partners to that of MTA1dE4 *	Interacting affinity and partners may alter due to Conformational changes *
	Effects on the aggressiveness of tumor cells	Migration	Promotion	Promotion (Higher activity)
		Invasion	Promotion	Promotion (Higher activity)
Clinical insight	Correlation with aggressive tumor characteristics (e.g., large tumor size and large number of tumors, etc.)		Yes [32,145]	Yes (Stronger) [32]
	Correlation with early HCC recurrence		Yes [32] (Studied at mRNA level)	Yes [32] (Studied at mRNA level) (A more sensitive marker)

* This information is based on our unpublished data.

## Data Availability

Data is contained within the article.

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
