# Peer review of "Molecular Mechanisms and Animal Models of HBV-Related Hepatocellular Carcinoma: With Emphasis on Metastatic Tumor Antigen 1"

_ijms, 2021, doi:10.3390/ijms22179380_

Round 1

Reviewer 1 Report

The manuscript is focused on the molecular mechanisms and animal models used in Hbv-related HCC therapy. Furthermore, the authors described the importance of using a suitable animal model for studying HBV-related hepatocarcinogenesis, that is the metastatic tumor antigen 1. The manuscript is important for the scientific network, but it is lacking of some therapeutic approaches currently under investigation for the HCC therapy, such as the virotherapy. Please refer to the review of Chianese A. et al (doi: 10.3390/cancers13112761) and improve the introduction section. The manuscript is well written and I approve it for publication after this modification. 

Author Response

Reviewer 1
Comments and Suggestions for Authors
The manuscript is focused on the molecular mechanisms and animal models used in HBV-related
HCC therapy. Furthermore, the authors described the importance of using a suitable animal model
for studying HBV-related hepatocarcinogenesis, that is the metastatic tumor antigen 1. The manuscript
is important for the scientific network, but it is lacking of some therapeutic approaches currently under
investigation for the HCC therapy, such as the virotherapy. Please refer to the review of Chianese A.
et al (doi: 10.3390/cancers13112761) and improve the introduction section. The manuscript is well
written and I approve it for publication after this modification.

Response:
The authors would like to thank the reviewer for the thorough and expert review of our manuscript.
We have edited the manuscript to reflect most of the suggestions from the reviewer. We have
summarized the current approaches to prevention of HBV-HCC (e.g., vaccination, antivirals and
chemotherapy as well as virotherapy) in the introduction section (lines 58-89). In addition, we also
added the potential therapeutic significance of MTA1 in subsection 4.4 section of the revised
manuscript (lines 599-623). In the revised manuscript, changes and edited texts were highlighted in
yellow.
Below are the changes and added texts in the revised manuscript:
- Introduction section of the revised manuscript:
Pages 2-3, Lines 58-89:
Using vaccination and anti-viral therapies to inhibit new infection and viral replication of HBV
is the first place to prevent the occurrence of HBV-HCC. These strategies are highly efficacious in
terms of interrupting the progression from chronic infection to HCC but do not eliminate the risk of
HCC development entirely[21-23]. Therefore, early detection of HCC in chronic hepatitis B (CHB)
patients, effective treatments and prevention of recurrence are of great importance for further
reducing the HBV infection-associated mortality. Currently, several predicting models incorporating
clinical features and viral factors (e.g., serum levels of HBV DNA, HBeAg, and HBsAg) have been
developed to evaluate the risk of HCC development in CHB patients [21]; and periodic screening
using ultrasonagraphy with AFP among high risk groups is the most frequently used strategy for
early detection of HBV-HCC. However, the sensitivity and specificity of these methods are not
satisfactory so far. Surgical resection is considered as the first-line therapeutic option for HCC in the
initial phase; nevertheless, only a few patients are candidates for surgery [24, 25] because most
patients are diagnosed at the advanced stage while few therapeutic options are available. Although
the advent of molecular-targeted drugs, such as multi-kinase inhibitor sorafenib, has advanced the
systemic therapies for advanced-stage patients, the survival benefit remains quite limited [26, 27];
and the long-term use of these drugs is frequently associated with toxicity and drug resistance [28,
29]. Furthermore, even in patients with resectable tumors, frequent tumor recurrence after surgery
remains the major cause of death for HBV-HCC. The heterogeneous molecular nature of HCC
affecting treatment outcome and recurrence is another reason for the poor prognosis of HCC [4].
These issues highlight the need of developing more effective approaches for early detection,
recurrence prediction and treatment of advanced HBV-HCC. Advancement of biomedical research
and technologies have greatly facilitated the identification of more gene expression signatures and
molecular mechanisms associated with HBV-HCC and made great progress in many aspects. Novel
therapeutic options, such as different molecular targeted therapies, epigenetic modulators,
immunotherapies, oncolytic virotherapies and even chemopreventive agents, alone or in
combination, are continuous emerging [30]. New biomarkers for early detection and prediction of
treatment outcomes are also under active investigation [31]. Employment of appropriate HBV-HCC
animal models that are relevant to human clinical settings are of paramount importance to
successfully translate basic scientific findings into clinical ap-plications for patients.
- Subsection 4.4 of the revised manuscript:
Page 15, Lines 599-623:
4.4 Therapeutic Significance of MTA1
Prognosis of patient with advanced HCC is poor partially due to therapeutic resistance to
traditional chemotherapeutic agents (e.g., sorafenib). Therapeutic resistance of cancer is an important
factor in contributing to the invasiveness and metastasis of cancer cells [169, 170], which relies on
escaping apoptosis and increasing drug efflux [171]. In recent years, several studies have focused on
MTA1 as an important mediator of cancer therapeutic resistance [172-176]. Despite the fact that the
role of MTA1 in therapeutic resistance has not been studied and discussed on HCC, these studies
provide information about emerging functions and mechanisms of MTA1 in aggressive cancers.
Moreover, MTA1 is gradually perceived to be a new drug target candidate for cancer treatment.
Recently, MTA1-targeted chemopreventive agents for cancer therapy are continuously developed
and studied [177-179]. For instance, natural compound resveratrol and its analog pterostilbene have
been recognized as the potent inhibitors for MTA1 and MTA1-guided signaling [180-182]. These
compounds appear to regulate the PTEN/AKT and P53 signaling pathways through inhibition of
MTA1/HDAC unit of the NuRD complex in cancer cells and finally inhibit tumor growth,
progression and metastasis of cancers. A clinical trial evaluating the therapeutic efficacy of
pterostilbene, an MTA1 inhibitor, for the treatment of prostate cancer has been designed and is
ongoing [Levenson, A.S. (2018) MTA1-targeted strategies for prostate cancer management.
Identification No. 9511118. Retrieved from https://reporter.nih.gov/projectdetails/
9511118#details]. Although this project is not conducted for HCC, it still offers clinical proof
for pterostilbene as a promising natural agent for MTA1-targeted chemopreventive and therapeutic
strategies to curb cancer. Together, these findings highlight the potential of using MTA1 as a
promising therapeutic target and facilitating MTA1-targeted strategies as well

Reviewer 2 Report

In this manuscript, the authors are trying to summarize molecular mechanisms and animal model of HBV-related HCC, and to introduce the significance of MTA1 in the development and progression of HBV-related HCC. The aim of this manuscript is interesting, and manuscript itself is well written overall. However, I found a number of concerns that need to be addressed.

General comment:
The sections 2 and 3 are interesting; however, because of that, there are a number of review articles specifying oncogenic mechanisms of HBV-induced hepatocarcinogenesis, and its animal models. Therefore, these sections should be shortened, and the authors might want to more focus on MTA1 and MTA1dE4 in HBV-related liver diseases. In the present manuscript, the authors described the expression and consequences of MTA1 in liver. However, molecular functions and diagnostic and therapeutic aspects of MTA1 and MTA1dE4 would increment our current knowledges in HBV-induced hepatocarcinogenesis.

Specific comments:

  1. Introduction: The authors should revise some sentences (for example, lines 37-51), because from which we might get the impression that they will summarize the general aspects of HCC. The authors should more focus on HBV, and describe HBV-related aspects of HCC more.

  2. Figure 1, lines 94-95: These parts should be revised. HBV genome produces five viral RNAs including two 3.5kb RNAs (precore and pregenomic RNA). Precore protein (p25) is produced from precore RNA while core protein is produced from pregenomic RNA. HBeAg is produced by the post-translational processing of precore protein, not of core (HBcAg) (Virology 2017;505:155-161). The subsection 2.1 and figure 1 seem not necessary as the content has been extensively reviewed elsewhere.

  3. 2.2 should be deleted or merged with the introduction of the Section 2. Likewise, some of the first parts of most subsections in the section 2 are redundant because they have been described in the introduction of the Section 2.

  4. The authors should summarize the contents of 2.3-2.6 in one figure.

  5. 2.4: c-MYC is also one of the important transcription factors for HBx-induced hepatocarcinogenesis (Int J Mol Sci 2019;20:5714).

  6. lines 273-281: Basically, as a consequence of immune tolerance, the product of a transgene does not induce inflammatory response in a transgenic mouse. Thus, it is necessary to explain how pre-S1/S2 protein produced from transgene induces inflammation and liver injury in mice.

  7. As an important mouse model for studying HBV, humanized mice with human liver should be noted, and please discuss its potential for studying HBV-related HCC.

  8. 4.3: The possible functions of MTA1dE4 and its difference from full-length MTA1 in the development and progression of HBV-related HCC should be discussed from a more detailed molecular viewpoint.

  9. Because this manuscript is a review article, Figures 3-4 are unnecessary. Instead, a figure summarizing putative functions of MTA1dE4 and/or MTA1.

  10. 4.5 and 4.6 require substantial revision as these parts should be described in the Section 3. The authors should not describ your current data; rather gather and summarize current knowledge based on recent findings regarding MTA1 and MTA1A4 in HBV-related HCC as pointed out in my comment #8.

Reviewer 3 Report

Thank you for the opportunity to evaluate this narrative review article on animal models of HBV-associated HCC.

While the authors have gone to considerable lengths to write a thorough review (the text is very long and more or less easy to follow), it does not appear that the authors applied a very rigorous methodology to present important findings much beyond the scope of their own previous work.

While other pre-clinical models exist for the study of HCC (organoids, cell lines, tumor explants, xenografts, etc.), the authors focus their discussion exclusively on animal models with HBV or other hepadnavirus involvement. They also dedicate a disproportionate amount of the text to discussing their own work on woodchuck hepatitis virus (WHV) and metastatic tumor antigen 1 (MTA1), a marker of invasive disease and poor prognosis. There are many other markers of poor prognosis in HCC, and it might be more helpful if the authors were to provide current understanding on early biomarkers that might allow for timely diagnosis and treatment before disease has devolved to an invasive state.

In all the discussion on HBV, no mention is made regarding the important roles that vaccines and antivirals have played in preventing HCC from developing in the context of HBV infection in the first place.

Author Response

Response to Reviewer 3:

The authors would like to thank the reviewer for the thorough and expert review of our manuscript. We have substantially edited and refined the manuscript to reflect most of the suggestions from the reviewer. In the revised manuscript, changes and edited texts were highlighted in yellow.

Point 1: While the authors have gone to considerable lengths to write a thorough review (the text is very long and more or less easy to follow), it does not appear that the authors applied a very rigorous methodology to present important findings much beyond the scope of their own previous work.

Response 1:

We thank the reviewer for the comment. In this manuscript, we would like to introduce the MTA1 as an example to address the importance of a suitable animal model that is critical for studying HBVrelated hepatocarcinogenesis. We thus edited the text of introduction section with more focus on HBVrelated aspects of HCC. In addition, we have refined the discussion about our own experience on using the HBV-HCC models and rearranged these contents to revised subsections 4.5-4.7 (lines 624-711) for a better presentation.

Below are the changes and added texts in the revised manuscript:

- Subsections 4.5-4.7 of the revised manuscript:

Pages 15-18, Lines 624-711: 4.5 Application of Hepadnavirus-Induced Woodchuck HCC Model for Studying the Biological

Functions and Clinical Significance of MTA1 in HBV-Related HCC

Our previous study [36]provided several lines of evidence to demonstrate that an HBV natural infection model, the woodchuck hepatitis virus (WHV)-infected woodchuck, is an appropriate model to study the role of MTA1 in hepadnavirus-induced HCC by characterizing the molecular function(s)

of woodchuck MTA1. Similar to human MTA1, woodchuck MTA1 (wk-MTA1) was overexpressed in WHV-induced HCC of the wood-chuck and played an indispensable role in the hepadnavirus X protein-mediated NF-κB activation and cell migration. In this study, we are the first to reveal that the expression of a major spliced variant of wk-MTA1, MTA1dE4 (exon 4–excluded form of

MTA1), but not the total wk-MTA1, was associated with more malignant characteristics of woodchuck HCC, including multiple tumors and a larger size. Using the woodchuck model, we can study the complex temporal relationships between MTA1/MTA1dE4 expression and the progression of HBV-related liver diseases including chronic hepatitis and HBV-related HCC. For instance, we could monitor the dynamic changes of target gene

expression (e.g., MTA1 and MTA1dE4) in the liver of woodchuck after experimental WHV infection within a reasonable time period (Figure 4). Moreover, we could assess the expression of the fulllength MTA1 and the spliced variant MTA1dE4 in both tumor and non-tumor tissues, all of which

were obtained by serial liver biopsy at different time points, and analyzed the correlation between them and the size of tumor as well as the result of serological assay (Figure 5). It can simultaneously implement various technologies to monitor the tumor take rate and growth rate in woodchucks as well as to correlate histopathological results at various time points with serum and biomarkers, which cannot be longitudinally determined in patients. By taking this advantage, it can offer a pre-clinical model for testing the promising MTA1-targeted chemopreventive agents and therapeutic strategies to curb HBV-related HCC.

4.6 Translational Researches Based on the Woodchuck Model

For decades, researchers have kept on exploring explore more sensitive biomarkers and developing more effective therapeutic strategies for HCC patients with a high risk of tumor recurrence. However, a great number of patients with HBV-HCC still suffer tumor recurrence [183, 184]. Our woodchuck HCC study demonstrated that MTA1dE4, a major spliced variant of MTA1, may represent a more sensitive marker than total MTA1 in WHV-induced HCC [35]. We thus analyzed the relationships between clinical characteristics of patients with HBV-HCC and expression of total MTA1 and MTA1dE4. We also explored the clinical impact of MTA1 and one of its major spliced variants, MTA1dE4, on postoperative recurrence in patients with HBV-related HCC in Taiwan through a 4-year retrospective cohort study [35]. For the first time, we revealed the clinical significance of MTA1dE4. It overexpressed in a higher percentage than total MTA1 in HBVHCC

patients and can serve as a more sensitive marker to predict the onset of early recurrence of HCC, especially in patients with low AFP. Therefore, we successfully translated the results of basic science research into human research and showed the potential clinical application of MTA1 in HBV-induced hepatocarcinogenesis. This is might represent a demonstration that new findings yielded by the woodchuck HCC model could be extrapolated and applied to HBV-related HCC.

4.7 Potential Issues on Application of HDT-Based Murine Models for Oncogenic Collaboration Studies in HBV-Related HCC. The flexibility of HDT technique in terms of transgenes and strains of the recipient mice renders

it the ideal approach to determine the in vivo oncogenic potential of the MTA1 and MTA1dE4. In this case, we established an HDT-based HCC mouse model in C57BL/6JNarl mice with coadministration of NrasV12/MYC oncogenes either with MTA1 or MTA1dE4 to assess their additive effect in terms of promoting tumorigenesis. Consistent with previous studies [124, 185], we found that hydrodynamic injection of NrasV12/MYC oncogenes induced the development of nodular and diffuse HCC within 8 weeks in some mice (Figure 6A). However, we observed that the variability of tumor burdens (including size and number) among mice in the same group is too large to investigate the additive effect of MTA1 or MTA1dE4 on accelerating tumorigenesis. We suggested that the variability in tumor burdens was mainly due to different expression levels of oncogenes, which were caused by the close-to-random insertions of transposons. Furthermore, we also speculate that the genetic background, in addition to the close-to-random

insertion effect, could be a possible factor affecting the rate of HCC formation in distinct mouse strains. We stably co-expressed NrasV12 and c-MYC oncogenes in the livers of different mouse strains (e.g., CBA/CaJNarl, C57BL/6JNarl and C3H/HeNCrNarl strains) with the same approach

and observed a dramatic divergence in both tumor latency and tumor burdens among these mouse strains was observed (Figure 6B). The possible explanation for this phenomenon awaits further investigation.

Point 2: While other pre-clinical models exist for the study of HCC (organoids, cell lines, tumor explants, xenografts, etc.), the authors focus their discussion exclusively on animal models with HBV or other hepadnavirus involvement.

Response 2:

We thank the reviewer for this suggestion. We have discussed the human liver-chimeric mouse model, an important model for studying HBV, which is established by the engraftment of human hepatocytes or clinical HBV-HCC specimens into the liver of an immunodeficient mouse. We accordingly summarized the current knowledge about the human liver-chimeric mice in studying HBV-related HCC and added the content in subsection 3.3 of the revised manuscript (lines 449-474).

Below are the added texts in the revised manuscript:

- Subsection 3.3 of the revised manuscript: Pages 10-11, Lines 449-474:

3.3 Human Liver-Chimeric Mice Model

Chimeric mice with humanized liver contain repopulated human hepatocytes in the majority of the liver. To establish engraftment, suitable recipient mice must be immunodeficient to prevent xenograft rejection, and they also exhibit endogenous liver damage to allow expansion of the transplanted hepatocytes. For more detailed information about the establishment of human liverchimeric mice please refer to review [97, 144, 145]. In contrast to genetically engineered mouse models, the human liver-chimeric mice are

susceptible to HBV infection and are capable of forming HBV cccDNA as well as supporting HBV replication. Considering chronic hepatitis B is a host-specific immune-mediated liver disease, one of the shortcomings of human liver-chimeric models is their highly immunodeficient background. Thus, in order to enable the study of HBV-related immunopathogenesis, the dual chimeric mouse models with simultaneous engraftment of both hepatocytes and human hematopoietic stem cells (HSCs) are gradually developed. However, the establishment of dual chimeric mice, particularly those with efficient reconstitution of the human immune system, may hamper by technical difficulties. Importantly, after HBV infection, the dually engrafted mice sustained high viremia, exhibited specific immune and inflammatory responses and showed the progression from chronic hepatitis to liver cirrhosis, but HCC development was not observed [145, 146]. Because HCC takes a long period of time to form, it may be difficult for dual chimeric mice to recapitulate such HBV-associated liver pathology. Nevertheless, this in vivo model still be helpful in terms of elucidating oncogenic pathways involving early phases of HCC initiation and progression. Engraftment of human liver cells expressing oncogenes or shRNAs against cellular RNAs and HBV DNA may help dissect the roles of specific genes or signaling pathways in HBV infection as well as HBV-related hepatocarcinogenesis. Accordingly, the dual chimeric mouse represents an idea small animal model for HBV-related research and maybe acutely required in the future.

Point 3: They also dedicate a disproportionate amount of the text to discussing their own work on woodchuck hepatitis virus (WHV) and metastatic tumor antigen 1 (MTA1), a marker of invasive disease and poor prognosis. There are many other markers of poor prognosis in HCC, and it might be more helpful if the authors were to provide current understanding on early biomarkers that might allow for timely diagnosis and treatment before disease has devolved to an invasive state.

Response 3:

We entirely agree with the reviewer’s comment about the need for discussion of the current markers for HBV-HCC prognosis. We thus added the description of the current marker of poor prognosis in HBV-HCC in section 4 of revised manuscript (lines 484-493).

Below are the added texts in the revised manuscript:

- Section 4 of the revised manuscript: Page 12, Lines 484-493:

  1. Relationship between The MTA1 and HBV-Related HCC

The lack of sensitive biomarkers for timely diagnosis and effective therapeutics for advanced tumors are two major reasons for the poor outcome of HCC, showing an imperative need to identify potential markers and therapeutic targets of HCC. The most well-studied HCC biomarkers are the alpha-fetoprotein (AFP), its isoform AFP-L3; and des-γ-carboxy prothrombin (DCP). However, the current commonly used markers, such as AFP, cannot effectively predict tumor recurrence. In an effort to identify novel markers for the detection of HCC recurrence, many other molecules have been explored, including glypican 3 (GPC3), cytokeratin 19 (CK19), glutamine synthase (GS), etc. For a more de-tailed information, please refer to the review of Zacharakis et al. [147]. The metastatic tumor antigen 1 (MTA1) is closely correlated to poor prognosis and frequent

postoperative recurrence in patients with HCC, particularly in those with HBV-HCC [148-150]. Thus, MTA1 has the potential to serve as a prognostic marker and therapeutic target for HBV-HCC. Most of the current understanding of the MTA1 functions in HCC and in other cancer types is derived from in vitro studies. However, the need for exploration of the clinicopathological significance and potential clinical applications of MTA1 cannot be satisfied completely with in vitro experiments. Suitable animal models are essential to advance our understanding of the underlying molecular and pathophysiological mechanisms with respect to MTA1 in HBV-HCC development and to develop new therapeutic strategies.

Point 4: In all the discussion on HBV, no mention is made regarding the important roles that vaccines and antivirals have played in preventing HCC from developing in the context of HBV infection in the first place.

Response 4:

We thank the reviewer for this suggestion. We have summarized the current approaches to prevention of HBV-HCC (e.g., vaccination, antivirals and chemotherapy as well as virotherapy) in the introduction section of revised manuscript (lines 56-72). In addition, we also added the potential

therapeutic significance of MTA1 in subsection 4.4 section of the revised manuscript (lines 599- 623).

Below are the added texts in the revised manuscript:

- Introduction section of the revised manuscript: Pages 2-3, Lines 58-89:

Using vaccination and anti-viral therapies to inhibit new infection and viral replication of HBV is the first place to prevent the occurrence of HBV-HCC. These strategies are highly efficacious in terms of interrupting the progression from chronic infection to HCC but do not eliminate the risk of

HCC development entirely[21-23]. Therefore, early detection of HCC in chronic hepatitis B (CHB) patients, effective treatments and prevention of recurrence are of great importance for further reducing the HBV infection-associated mortality. Currently, several predicting models incorporating

clinical features and viral factors (e.g., serum levels of HBV DNA, HBeAg, and HBsAg) have been developed to evaluate the risk of HCC development in CHB patients [21]; and periodic screening using ultrasonagraphy with AFP among high risk groups is the most frequently used strategy for early detection of HBV-HCC. However, the sensitivity and specificity of these methods are not satisfactory so far. Surgical resection is considered as the first-line therapeutic option for HCC in the initial phase; nevertheless, only a few patients are candidates for surgery [24, 25] because most patients are diagnosed at the advanced stage while few therapeutic options are available. Although the advent of molecular-targeted drugs, such as multi-kinase inhibitor sorafenib, has advanced the systemic therapies for advanced-stage patients, the survival benefit remains quite limited [26, 27];

and the long-term use of these drugs is frequently associated with toxicity and drug resistance [28, 29]. Furthermore, even in patients with resectable tumors, frequent tumor recurrence after surgery remains the major cause of death for HBV-HCC. The heterogeneous molecular nature of HCC

affecting treatment outcome and recurrence is another reason for the poor prognosis of HCC [4]. These issues highlight the need of developing more effective approaches for early detection, recurrence prediction and treatment of advanced HBV-HCC. Advancement of biomedical research

and technologies have greatly facilitated the identification of more gene expression signatures and molecular mechanisms associated with HBV-HCC and made great progress in many aspects. Novel therapeutic options, such as different molecular targeted therapies, epigenetic modulators, immunotherapies, oncolytic virotherapies and even chemopreventive agents, alone or in combination, are continuous emerging [30]. New biomarkers for early detection and prediction of treatment outcomes are also under active investigation [31]. Employment of appropriate HBV-HCC animal models that are relevant to human clinical settings are of paramount importance to

successfully translate basic scientific findings into clinical ap-plications for patients.

- Subsection 4.4 of the revised manuscript: Page 15, Lines 599-623:

4.4 Therapeutic Significance of MTA1

Prognosis of patient with advanced HCC is poor partially due to therapeutic resistance to traditional chemotherapeutic agents (e.g., sorafenib). Therapeutic resistance of cancer is an important factor in contributing to the invasiveness and metastasis of cancer cells [169, 170], which relies on

escaping apoptosis and increasing drug efflux [171]. In recent years, several studies have focused on MTA1 as an important mediator of cancer therapeutic resistance [172-176]. Despite the fact that the role of MTA1 in therapeutic resistance has not been studied and discussed on HCC, these studies provide information about emerging functions and mechanisms of MTA1 in aggressive cancers. Moreover, MTA1 is gradually perceived to be a new drug target candidate for cancer treatment. Recently, MTA1-targeted chemopreventive agents for cancer therapy are continuously developed

and studied [177-179]. For instance, natural compound resveratrol and its analog pterostilbene have been recognized as the potent inhibitors for MTA1 and MTA1-guided signaling [180-182]. These compounds appear to regulate the PTEN/AKT and P53 signaling pathways through inhibition of

MTA1/HDAC unit of the NuRD complex in cancer cells and finally inhibit tumor growth, progression and metastasis of cancers. A clinical trial evaluating the therapeutic efficacy of pterostilbene, an MTA1 inhibitor, for the treatment of prostate cancer has been designed and is ongoing [Levenson, A.S. (2018) MTA1-targeted strategies for prostate cancer management. Identification No. 9511118. Retrieved from https://reporter.nih.gov/projectdetails/9511118#details]. Although this project is not conducted for HCC, it still offers clinical proof for pterostilbene as a promising natural agent for MTA1-targeted chemopreventive and therapeutic strategies to curb cancer. Together, these findings highlight the potential of using MTA1 as a promising therapeutic target and facilitating MTA1-targeted strategies as well as developing combinatorial therapeutic approaches for cancer treatment in the future.

Round 2

Reviewer 1 Report

The revised version is recommended for the publication. 

Reviewer 2 Report

I appreciate the efforts that the authors had made to address my comments.

The manuscript is now acceptable for publication.

Reviewer 3 Report

Thank you for the changes that have been made.